# Electrospun Nanofiber Dopped with TiO_2_ and Carbon Quantum Dots for the Photocatalytic Degradation of Antibiotics

**DOI:** 10.3390/polym16212960

**Published:** 2024-10-22

**Authors:** Valentina Silva, Diana L. D. Lima, Etelvina de Matos Gomes, Bernardo Almeida, Vânia Calisto, Rosa M. F. Baptista, Goreti Pereira

**Affiliations:** 1Department of Chemistry, Centre for Environmental and Marine Studies (CESAM), University of Aveiro, 3810-193 Aveiro, Portugal; valentinagsilva@ua.pt (V.S.); vania.calisto@ua.pt (V.C.); 2H&TRC—Health & Technology Research Center, Coimbra Health School, Polytechnic University of Coimbra, Rua 5 de Outubro, 3045-043 Coimbra, Portugal; diana.lima@estesc.ipc.pt; 3Centre of Physics of Minho and Porto Universities (CF-UM-UP), Laboratory for Materials and Emergent Technologies (LAPMET), University of Minho, 4710-057 Braga, Portugal; emg@fisica.uminho.pt (E.d.M.G.); bernardo@fisica.uminho.pt (B.A.)

**Keywords:** water treatment, pharmaceuticals, electrospinning, photocatalysis, amoxicillin, sulfadiazine

## Abstract

Novel photocatalysts were synthesized through the association of carbon quantum dots (CQDs) with commercial (P25) titanium dioxide (TiO_2_) by sonication. The resulting TiO_2_/CQDs composite was then incorporated into the polyamide 66 (PA66) biopolymer nanofibers using the electrospinning technique, considering a composite nanoparticles-to-polymer ratio of 1:2 in the electrospinning precursor solution. The produced nanofibers presented suitable morphology and were tested for the photocatalytic degradation under simulated solar radiation of 10 mg L^−1^ of amoxicillin (AMX) and sulfadiazine (SDZ), in phosphate buffer solution (pH 8.06) and river water, using 1.5 g L^−1^ of photocatalyst. The presence of the photocatalyst increased the removal of AMX in phosphate buffer solution by 30 times, reducing the AMX degradation half-life time from 62 ± 1 h (without catalyst) to 1.98 ± 0.06 h. Moreover, SDZ degradation half-life time in phosphate buffer solution was reduced from 5.4 ± 0.1 h (without catalyst) to 1.87 ± 0.05 h in the presence of the photocatalyst. Furthermore, the PA66/TiO_2_/CQDs were also efficient in river water samples and maintained their performance in at least three cycles of SDZ photodegradation in river water. The presented results evidence that the produced photocatalyst can be a promising and sustainable solution for antibiotics’ efficient removal from water.

## 1. Introduction

Wastewater treatment plants are inefficient for removing antibiotics, so developing more effective methods for eliminating these contaminants from aqueous effluents is essential to reduce their entrance into natural waters. One of the most consumed antibiotics worldwide is amoxicillin (AMX), from the penicillin family, which has been included in the last watch lists published in the context of the European Council Water Framework Directive (3rd watch list (Decision 2020/1161) and 4th (Decision 2022/1307)). AMX is often detected in the aquatic environment due to the continuous disposal of contaminated effluent, which could increase antimicrobial resistance. Sulfadiazine (SDZ), a sulfonamide, is also a widely used broad-spectrum antibiotic, utilized for treating or preventing bacterial infections in both humans and livestock due to its therapeutic efficacy and affordable cost [1,2]. SDZ and AMX’s chemical stability and nonbiodegradability promote their persistence in aquatic environments over a long period, impacting human health, being toxic to several organisms, and inhibiting the nitrification process in activated sludge systems [3,4,5,6]. The concentrations of SDZ and AMX found in surface and groundwater can range from ng L^−1^ to μg L^−1^, while in wastewater increase up to mg L^−1^ [2,7,8]. Thus, their persistence in the environment may result in antimicrobial resistance and organism distress, causing a serious environmental issue [9].

The application of solar-driven photocatalysis, an advanced oxidation process using a recoverable and reusable photocatalyst, could be a sustainable alternative to efficiently remove antibiotics from water resources. The semiconductor titanium (IV) dioxide (TiO_2_) is among the most used photocatalysts due to its advantages, such as strong oxidizing power, commercial availability, economic feasibility, non-toxicity, and biological and chemical inertness [10,11]. However, TiO_2_-based photocatalysts are associated with some drawbacks: (i) large band gap energy, restricting the use of TiO_2_ primarily with ultraviolet (UV) light; (ii) rapid electron–hole recombination; and (iii) difficult recovery from aqueous systems (where ultrafiltration, for example, is required), making it difficult to reuse and increasing the cost of the treatment process [12].

Carbon quantum dots (CQDs) are zero-dimensional carbon-based semiconductor nanoparticles, possessing several advantages such as biocompatibility, low toxicity, high photostability, strong absorption band, chemical stability, and photoluminescence (PL). These carbon spheres (or quasi-spherical) have predominantly sp^2^ carbon fused by diamond-like sp^3^-hybridized carbon intersections [13,14,15]. In recent years, CQDs have emerged as an efficient approach to enhance the catalytic performance of photocatalysts, especially TiO_2_, under visible light [16,17]. The association of CQDs with TiO_2_ can improve the catalytic performance due to CQDs’ suitable band gap, exceptional electron donor/acceptor properties, and exceptional electron transfer features [12,14,18,19]. 

The immobilization of the photocatalyst has been seen as an alternative to promote easy recovery and reuse from treated water. Polymers such as polyvinylidene fluoride (PVDF), poly(vinyl pyrrolidone) (PVP), polyacrylonitrile (PAN), poly(lactic acid) (PLA), polyether sulfone (PES), and polyamide-6 (PA-6) have been used as polymer matrices to support TiO_2_-based photocatalysts [20,21]. Within the methods used for immobilizing photocatalysts in polymeric matrices, electrospinning has gained substantial attention due to its effectiveness and versatility [20], producing nanofibers with a high surface area and porosity [22,23].

The nanofibers/TiO_2_ composites have shown great results in the removal of different pollutants, such as organic dyes (methyl orange, rhodamine B, and methylene blue) and antibiotics (sulfamethoxazole and enrofloxacin) [24,25,26]. Deals et al. [27] incorporated TiO_2_ nanoparticles in polyamide 6 (PA6) fibers by electrospinning and observed a good photocatalytic performance in the degradation of methylene blue (MB). However, in this study, UV light irradiation was used and the photocatalyst reuse was not attempted. In order to use visible light, Almeida et al. [24] prepared TiO_2_ and graphene oxide (GO) composites, and incorporated them in poly(vinylidene difluoride-co-trifluoroethylene) (P(VDF-TrFE)) fibers for the photodegradation of MB. Despite the good photocatalytic results, the recovery and reuse of the catalyst were not performed. Recently, Lin et al. [25] incorporated TiO_2_/GO composites in PAN membranes for the photodegradation of two antibiotics (5 mg L^−1^), sulfamethoxazole (SMX) and enrofloxacin (ENR). Under visible light irradiation, TiO_2_/GO/PAN enhanced the removal of these antibiotics, achieving a total removal after 180 min, with approximately 50% attributed to adsorption. Furthermore, this TiO_2_/GO/PAN composite maintained a good performance after five cycles, although requiring a washing step after each use.

In this work, a new photocatalyst was prepared by incorporating a TiO_2_/CQDs composite in polyamide 66 (PA66) nanofibers by electrospinning, and applied in the removal of antibiotics from different aqueous matrices. In previous studies, TiO_2_/CQDs composites showed excellent photocatalytic efficiency, under solar light irradiation, for the degradation of antibiotics from water [19,28], but evidencing some challenges concerning their reusability. Electrospun PA66 nanofibers were selected for this work to be combined with TiO_2_/CQDs due to their insolubility in water, good chemical and thermal resistance, high mechanical strength, and low cost [29,30], allowing to obtain a novel and reusable photocatalyst which presented great efficiency under solar light irradiation. Herein, we demonstrated the photocatalytic efficiency of PA66/TiO_2_/CQDs nanofibers in removing two antibiotics, AMX and SDZ, from different water matrices. Furthermore, as proof of concept, the recovery and reusability of the polymeric photocatalyst were evaluated in river water for SDZ.

## 2. Experimental Section

### 2.1. Synthesis of Carbon Quantum Dots (CQDs)

CQDs with citric acid and urea were prepared by a hydrothermal treatment adapted from Silva et al. [31]. Briefly, citric acid (3.0 g) and urea (1.0 g) were mixed with 10 mL of ultrapure water and placed into a 70 mL autoclave for 5 h at 180 °C; then, the largest particles were removed by centrifugation at 5000 rpm for 30 min, using a centrifuge (SIGMA 4–10, B. Braun, Melsungen, Germany) and the remaining solution was purified by cycles (5–7) of precipitation with propan-2-ol and centrifugation at 5000 rpm for 10 min. When the CQDs were aggregated at the bottom, the remaining propan-2-ol was removed and the CQDs were dried at 50 °C.

### 2.2. Synthesis of TiO_2_/CQDs Composite

The TiO_2_/CQDs composite was synthesized by an easy sonication methodology. Briefly, 1.00 g of TiO_2_ powder was dispersed in 30 mL of ethanol in a flask in an ultrasonic bath at 70 °C for 5 min. Then, 0.833 mL of an aqueous solution of 50 g L^−1^ of CQDs was added and left to react for 6 h at 70 °C in the ultrasonic bath to obtain the composite with 4% (*w*/*w*) of CQDs in TiO_2_. Then, the composite was dried at 75 °C.

### 2.3. Preparation of PA66 Nanofibers Containing TiO_2_/CQDs by Electrospinning

The preparation of the polymer solution involved dissolving 0.5 g of PA66 (polyamide 66 or nylon 66) in 5 mL of 1,1,1,3,3,3-Hexafluoroisopropanol (HFIP) at room temperature and stirring at 500 rpm until complete dissolution. TiO_2_/CQDs (0.25 g) were then reduced to powder in a mortar, sieved through a granulometric sieve with a pore size of 40 µm, and added to the polymer solution. After ultrasonication for 10 min to reduce nanoparticle aggregates, the mixture was stirred vigorously to ensure uniform dispersion. For electrospinning (Figure 1A), the polymer suspension with TiO_2_/CQDs was introduced into a syringe fitted with an 18G needle at a constant flow rate of 2 mL/h applied to the pump. An electric field of 20 kV was applied between the needle and the collector, placed precisely 10 cm apart, to facilitate the formation of uniform polymer fibers on the surface of the collector.

By carefully controlling these electrospinning parameters, a uniform and controlled deposition of polymeric fibers containing the composite (PA66/TiO_2_/CQDs) on the collector surface was achieved. The nanofibers were prepared using the ratio of 1:2 TiO_2_/CQDs:PA66 (*w*/*w*).

As a control, PA66 nanofibers without the TiO_2_/CQDs composite were also prepared using the same electrospinning parameters.

### 2.4. Nanofibers Characterization

The morphology, size, and composition of the produced fibers were studied by scanning electron microscopy (SEM) and energy-dispersive X-Ray spectroscopy (EDS), using a scanning electron microscope (model SU70, Hitachi, Tokyo, Japan) operating at an accelerating voltage of 2 kV. Before SEM analysis, a conductive carbon thin film was deposited onto the fibers using a carbon rod coater (Emitech K950X, Quorum Technologies, Laughton, UK). The diameter range of fibers was measured by SEM images using the ImageJ 1.53k analysis software (NIH, https://imagej.nih.gov/ij/, accessed on 21 July 2021). The average diameter and diameter distribution were determined by measuring 120–140 random fibers from the SEM images.

The crystallinity of the nanofibers was analyzed by X-Ray diffraction (XRD) using a Panalytical X’Pert Pro3 diffractometer (Malvern Panalytical, Worcestershire, UK) with a Cu anode (K-α 1.54060 Å wavelength) handled at 45 kV and 40 mA in a 2θ range of 3–70°, at 25 °C.

Fourier transform infrared with attenuated total reflectance (FTIR-ATR) spectroscopy (Avatar 360 Nicolet spectrometer, Thermo Scientific, Waltham, MA, USA) was used to evaluate the chemical composition of PA66 and PA66/TiO_2_/CQDs nanofibers. For this, each sample was placed on the ATR window and scanned in the 500–4000 cm^−1^ range, with a resolution of 2 cm^−1^ in transmittance mode, and presented as an average of 64 readings.

### 2.5. Water Matrices Characterization

For the photocatalysis experiments, two water matrices were used in this study, namely: phosphate buffer solution (PBS, pH 8.06) and river water. River water was collected between May and June of 2024 in Douro River at Marina Santiago Melres (Gondomar, Portugal, pH 7.08) (41°04′21.8″ N, 8°24′53.9″ W), filtered through 0.45 μm nitrocellulose membrane filters (Millipore, Burlington, VT, USA), stored in the dark at 4 °C, and used for no longer than 15 days after collection. Both water matrices were characterized by measuring pH, salinity (PSU), conductivity (µS/cm), total dissolved solids (TDS), and dissolved oxygen (DO, %), using a Multi 3320 m from WTW.

Dissolved organic carbon was also measured using a total organic carbon analyzer, TOC-VCPH, from Shimadzu (Kyoto, Japan). For that purpose, a stock solution of 1000 mg L^−1^ potassium hydrogen phthalate (KHC_8_H_4_O_4_) was prepared in ultrapure water. The calibration curve was performed using standard solutions of KHC_8_H_4_O_4_ in ultrapure water (0.0–5.0 mg L^−1^), by dilution of proper amounts of the stock solution. The coefficient of determination (R^2^) and limit of detection (LOD) for the calibration curve obtained were 0.9991 and 0.34 mg L^−1^, respectively. Samples were filtered using a syringe filter of PVDF 0.22 μm (Whatman), acidified with 2% (*v*/*v*) of HCl 2 mol/L, and covered with Parafilm M^®^, previously to the analysis. The water matrices’ characterization values can be found in Table 1.

### 2.6. Photocatalytic Experiments

As described elsewhere [19,31], irradiation experiments were carried out in a solar radiation simulator Solarbox 1500 (Co.fo.me.gra, Milan, Italy) equipped with a xenon arc lamp (1500 W) and UV filters that limit the transmission of light below 290 nm. All experiments were performed with a constant irradiation of 55 W m^−2^ (290–400 nm), which corresponds to 550 W m^−2^ in the spectral range, according to the manufacturer. The level of irradiance and temperature was monitored by a multimeter (Co.fo.me.gra, Milan, Italy) equipped with a UV 290–400 nm band sensor and a black standard temperature sensor. A parabolic reflection system was used to ensure irradiation uniformity in the chamber, which was kept refrigerated by an air-cooling system.

For the kinetic photodegradation curves without the catalyst (photolysis), antibiotic aqueous solutions (20 mL, 10 mg L^−1^) were added to the quartz tubes (internal diameter × height = 1.8 × 20 cm), in triplicate, and irradiated at different times. Aliquots (1.0 mL) of replicates and dark controls were withdrawn at pre-set irradiation times (t, h), stored in the dark at 4 °C, and analyzed within 24 h for the antibiotic concentration (details are given in Section 2.7). Dark controls correspond to a quartz tube covered by aluminum foil maintained inside the solar simulator during the same periods as the irradiated samples, being exposed to identical conditions except for the irradiation.

As for the photocatalytic experiments, the irradiation was carried out under the same conditions, but in the presence of the photocatalyst (PA66/TiO_2_/CQDs, 1.5 g L^−1^, Figure 1B). For this, approximately 30 mg of the nanofiber was cut and added to the antibiotic aqueous solutions (10 mg L^−1^) in quartz tubes (internal diameter × height = 1.8 × 20 cm), in triplicate, and the mixture was irradiated at different times. The volume of the antibiotic was calculated considering the fiber mass, in order to have 1.5 g L^−1^ of the photocatalyst. Each set of experiments was accompanied by dark controls under identical conditions except for irradiation (quartz tubes covered by aluminum foil), maintained inside the solar simulator during the same time as the irradiated solutions. Aliquots (1.0 mL) of replicates and dark controls were withdrawn at pre-set irradiation times (t, h), stored in the dark at 4 °C and analyzed within 24 h for the antibiotic concentration (details are given in Section 2.7. Chromatographic analyses). The same experiments were carried out in the presence of the PA66 fibers (1.5 g L^−1^), to evaluate the contribution of the fibers in the removal of the antibiotics under study.

For the reuse experiments, the nanofiber was added to SDZ in river water (10 mg L^−1^), maintaining the photocatalyst dose at 1.5 g L^−1^, in quartz tubes, and irradiated for 4 h. Then, the antibiotic solution was removed from the quartz tube, and the same volume of a new SDZ solution was added. This procedure was repeated to perform three consecutive cycles. Each cycle was performed in triplicate, and dark controls were also evaluated. Each sample was filtered and analyzed according to Section 2.7.

The remaining concentration of antibiotic in irradiated solutions (*C*) was compared with that in the respective dark control (*C*_0_) for determining the percentage of degradation at each irradiation time (*t*, h). GraphPad Prism 8 was used to determine the fittings of experimental data to the pseudo first-order kinetic equation *C*/*C*_0_ = $e^{-kt}$, where *k* is the pseudo first-order degradation rate constant (h^−1^). Also, the antibiotics’ half-life times (t_1/2_) were calculated as $\frac{ln(2)}{k}$. GraphPad Prism 8 was used to apply a one-way ANOVA to compare the photocatalytic efficiencies of each reuse cycle.

### 2.7. Chromatographic Analysis

Quantitative analysis of the antibiotics in the aqueous phase was performed using high-performance liquid chromatography (HPLC). The device consisted of a Waters Alliance 2695 Separations Module equipped with a Waters 2487 Dual Absorbance detector. Separation was carried out using a 150 mm × 4.6 mm i.d. ACE^®^ C18 column-PFP (5 μm particle size) connected to a 4.6 mm i.d. ACE^®^ 5 C18 guard column at 25 °C. The mobile phase consisted of methanol: 0.1% formic acid, 20:80 (*v*/*v*) for both AMX and SDZ, at a flow rate of 0.8 mL min^−1^. Before use, the mobile phase was filtered through a 0.2 μm polyamide membrane filter (Whatman, Maidstone, UK). Samples and standards were filtered by a syringe filter of PTFE 0.22 μm (Labfil^®^, Alwsci, Shaoxing, China). The volume of injection was 60 μL for AMX and 20 μL for SDZ, while the detection of AMX and SDZ was performed at 230 nm and 270 nm, respectively. In order to obtain the calibration curve for each matrix, the corresponding standard solutions, with concentrations between 0.1 and 10 mg L^−1^, were prepared from the stock solutions in PBS (10 mg L^−1^) and analyzed in triplicate. The linear regression equations for each antibiotic were obtained and the respective limits of detection (LOD) and quantification (LOQ), in mg L^−1^, were determined by [LOD] = $\frac{\left(3\times S_{b}\right)}{m}$ and [LOQ] = $\frac{\left(10\times S_{b}\right)}{m},$ where *S_b_* is the standard deviation of the *y*-interception and *m* is the slope. An LOD of 0.1 and 0.05 mg L^−1^ was obtained for AMX and SDZ, respectively, and an LOQ of 0.3 and 0.2 mg L^−1^ for AMX and SDZ, respectively.

## 3. Results and Discussion

### 3.1. Morphological and Structural Analysis of PA66/TiO_2_/CQDs Nanofibers 

The TiO_2_/CQDs composites here presented have already shown great results in photocatalytic studies using solar radiation. In previous works, the authors observed that the association of some CQDs (Appendix A) with TiO_2_ P25 could decreased the semiconductor band gap, observing changings from 3.2 to 3.0 eV, for TiO_2_ and TiO_2_/CQDs, respectively [31].

During the electrospinning process, the flow of the polymer solution at the tip of the needle was steady, and there were no current fluctuations. The resulting fibers were white in appearance and exhibited a high degree of flexibility. For the PA66 fibers, SEM images (Figure 2A,B) presented homogeneous fibers with an average diameter of 253 ± 55 nm, and no beads or crystallites were observed on the surface of the fabricated fibers. However, SEM images of PA66/TiO_2_/CQDs (Figure 2C,D) revealed that the fibers exhibited increased roughness and greater heterogeneity, with diameters ranging around 300 ± 82 nm, indicating the incorporation of TiO_2_/CQDs into the fibers.

EDS analysis (Figure 3) confirmed the presence of Ti in the PA66/TiO_2_/CQDs fibers, distributed uniformly throughout the polymer, corroborating the uniform distribution of TiO_2_/CQDs in the fibers.

The X-Ray diffractogram of PA66/TiO_2_/CQDs nanofibers showed a high similarity with the TiO_2_/CQDs and TiO_2_ XRD pattern (Figure 4), indicating the successful incorporation of the TiO_2_/CQDs composite into the PA66 fiber. All XRD patterns showed the presence of TiO_2_ anatase and rutile crystalline phases. The XRD peaks at 2θ = 25.5°, 38.0°, 48.3°, 55.3°, 62.9°, and 70.4°, correspond to the anatase (101), (004), (200), (211), (204), and (220) crystal planes, respectively. The rutile pattern can be observed in the reflections at 2θ = 27.6°, 36.3°, 41.4°, and 54.1°, respectively corresponding to (110), (101), (111), and (211) crystal planes. These results are in accordance with the previous works and with the standard pattern (Joint Committee on Powder Diffraction Standards (JCPDS)) for anatase TiO_2_ (card No. 00-021-1272) and rutile TiO_2_ (card No. 00-021-1276) [32,33]. Moreover, the broad peaks at 2θ angles of 20.9° and 23.5° of the PA66/TiO_2_/CQDs nanofibers could be attributed to the presence of the PA66 polymer [34,35]. Moreover, the crystalline size of TiO_2_ in the PA66/TiO_2_/CQDs nanofibers was estimated as 20.6 nm by the Scherrer equation, which is similar to bare TiO_2_ (19.0 nm) and to TiO_2_/CQDs composites (20.3 nm).

FTIR-ATR spectra of PA66 and PA66/TiO_2_/CQDs (Figure 5) showed the presence of the same peaks at 3296 and 3084 cm^−1^ corresponding to the amino group stretching vibrations, 2932 and 2860 cm^−1^ from stretching vibrations of CH_2_, 1634 cm^−1^ attributed to the stretching vibrations of the amide carbonyl group, 1538 cm^−1^ from bending and stretching vibrations of -NH, and 1464 cm^−1^ corresponding to the stretching vibrations of CH_2_ [36,37]. Furthermore, the PA66/TiO_2_/CQDs presented a broad peak at 666 cm^−1^ that can be attributed to the stretching of Ti–O bonds [31]. These results indicated that the composite was incorporated into the fibers without affecting the polymer formation.

### 3.2. Photocatalytic Experiments

In previous works, TiO_2_/CQDs composites (500 mg L^−1^) evidenced good performance as photocatalysts in the solar-driven removal of antibiotics from water resources [19,31]. As mentioned before, these composites have two main setbacks: recovery from treated water and reuse. Therefore, to address these disadvantages, the TiO_2_/CQDs composite here produced was incorporated in nanofibers of PA66 with a ratio of 1:2 TiO_2_/CQDs:PA66 (*w/w*). To maintain the photocatalyst dosage used in previous works (0.5 g L^−1^ of TiO_2_/CQDs) [19,31], 1.5 g L^−1^ of PA66/TiO_2_/CQDs was used for the photocatalytic experiments.

For each antibiotic, photodegradation kinetics studies (Figure 6) were carried out (i) in the absence of the photocatalyst (photolysis), (ii) with bare PA66 fibers, and (iii) with PA66/TiO_2_/CQDs photocatalyst. The kinetics parameters [rate constants (*k*, h^−1^), determination coefficient (*R*^2^), and half-life times (*t*_1/2_, h)] for photolysis and photocatalytic experiments carried out in PBS and river water, for both antibiotics, were obtained by applying pseudo-first-order equation fittings (Table 2).

For AMX, initially, the comparison between photolysis, bare fiber PA66, and PA66/TiO_2_/CQDs photocatalyst was performed in PBS. The results showed that the bare fiber PA66 and the photolysis had a similar behavior (Figure 6A), showing almost no AMX removal after 6 h of irradiation, demonstrating that the removal of AMX was not impacted by the polymer. The rate of removal of AMX greatly increased with the incorporation of TiO_2_/CQDs in the PA66 fibers (Table 2 and Figure 6A). The application of PA66/TiO_2_/CQDs was able to remove AMX from water much faster than only photolysis, increasing the kinetic rate by around 30 and 17 times in PBS and river water, respectively.

In fact, the *t*_1/2_ decreased from 60 ± 1 h (in the absence of the photocatalyst) to 1.98 ± 0.06 h when the PA66/TiO_2_/CQDs photocatalyst was applied in PBS (pH 8.06). Furthermore, the results showed a good AMX removal even in river water, reducing the t_1/2_ for the AMX removal to 3.52 ± 0.09 h. In this case, the photodegradation of AMX in river water was lower when compared to PBS, which can be attributed to the matrix complexity or the pH. River water presented a higher content of organic matter than PBS (Table 1 shows higher values of TOC than PBS), thus, the photocatalyst could be used to degrade organic matter present in this matrix at the same time as AMX, having a competitive behavior in the photodegradation. Moreover, the literature reports that the photodegradation of AMX is influenced by the matrix pH, being faster at higher pH values [6].

A control was performed during the photocatalysis experiments, where the same experimental conditions were applied, but the tubes were maintained in complete darkness. These controls confirmed that all the antibiotic removal can be attributed only to photodegradation, excluding other processes like adsorption and thermal decomposition.

As for AMX, the SDZ removal was studied for all the different conditions in PBS, and then, the photocatalyst performance was tested in river water (Figure 6B and Table 2). The bare fiber PA66 showed a slower kinetic rate of photodegradation of SDZ than photolysis, indicating that the PA66 fiber exerts a filter effect that inhibits the photodegradation of SDZ. As happens with many antibiotics, depending on their structure and class, their interactions with the matrix, adsorbents, and photocatalysts differ. In this work, for SDZ, a sulphonamide, it was verified that the presence of bare fiber PA66 without composite exerts a negative effect in degradation of this antibiotic, decreasing its SDZ removal from solution comparatively to SDZ photolysis. The same was not verified for AMX, an aminopenicillin, whose removal had similar behavior with and without bare fiber PA66. Thus, for SDZ, the fiber has a filter effect, acting as scavenger of reactive species or quenching the excited states, and affecting the photodegradation process [6,38]. However, the application of the PA66/TiO_2_/CQDs photocatalyst was not affected by this phenomenon, as verified by a significant increase in the removal of SDZ. In fact, the *t*_1/2_ of SDZ decreased from 5.4 ± 0.1 h to 1.87 ± 0.05 h with the application of the photocatalyst in PBS. SDZ removal was faster in river water than in PBS, with a *t*_1/2_ decrease from 1.87 ± 0.05 h to 1.33 ± 0.05 h. In previous works, the photodegradation of SDZ also occurred at a higher rate in freshwater (pH 7.3) than in ultrapure water (pH 7.3), where pH had no influence. In fact, it was reported that the increase in pH from 6.3 to 7.3 has a positive influence on SDZ photodegradation due to a higher prevalence of SDZ in its negatively charged form [38]. In our case, the pH was not the same; however, the matrix where the photodegradation rate was higher was at a lower pH, indicating that the main factor affecting our degradation rate was the aqueous organic matter content that overcame the negative influence of the pH. Louros et al. [38] reported that the principal factor that impacted the high removal of SDZ in freshwater was the creation of excited triplet states of dissolved organic matter (^3^DOM*). As for the results of the dark controls of SDZ, it was confirmed that all the observed removal can be attributed to photodegradation, as was found for AMX.

Table 3 presents some studies reported in the literature regarding the photocatalytic degradation of AMX and SDZ using TiO_2_-based catalysts and simulated solar light irradiation. It is important to observe that a direct comparison between these photocatalysts is complex since different parameters were used, such as antibiotic concentration and matrix pH. For example, Gao et al. [39] obtained one efficient photocatalyst for the degradation of AMX (5 mg L^−1^) in ultrapure water, with a constant rate of 0.0614 min^−1^, preparing a composite based on TiO_2_, graphite carbon nitride (g-C_3_N_4_), and silver nanoparticles. Meanwhile, Thi et al. [40] coupled TiO_2_ to ZnO nanoparticles to remove AMX (50 mg L^−1^) by photocatalysis in the presence of O_3_, obtaining a *k* ranging from 0.0032 to 0.0198 min^−1^ when the pH varies from 3 to 11. Regarding SDZ removal, Louros et al. [28] prepared TiO_2_/CQDs composites (by an in situ method), obtaining a rate constant of 0.71 ± 0.02 h^−1^ for the photocatalytic removal of this antibiotic (10 mg L^−1^) in PBS (pH 8.3), under solar irradiation. Also, Silva et al. [19] prepared similar composites using TiO_2_ P25, achieving better photocatalytic performance with a *k* of 4.81 ± 0.06 h^−1^, in PBS pH 8.6.

When TiO_2_-based composites are used, the photocatalytic process initiates with the absorption of a photon by the semiconductor, exciting an electron (e^−^) to the conduction band and generating a hole (h^+^) in the valence band. These two charged species can recombine and transfer heat or participate in redox reactions. From the reports found in the literature, photocatalysis with TiO_2_-based composites occurred mainly by redox reactions with water molecules, forming reactive oxygen species (ROS). The formed ROS will then react with the organic pollutants, degrading them [41,42]. Given the efficiency of the evaluated composites, future studies on the specific mechanism of the photocatalytic degradation of AMX and SDZ by this photocatalyst should be conducted, including the evaluation of photodegradation byproducts and their toxicity.

As a proof of concept, some recovery and reuse studies were performed, using PA66/TiO_2_/CQDs in subsequent cycles for the removal of SDZ from river water (Figure 7). The results showed a slight enhancement in SDZ removal until the 3rd cycle of reuse, increasing from 84% (1st cycle) to 88% (3rd cycle). One-way ANOVA revealed that only the photocatalytic results corresponding to cycles 1 and 3 are significantly different (*p* < 0.05), with dark controls presenting no significant difference between the three cycles. The small increase in efficiency of the photocatalyst with the reuse cycles can be due to the increase in the wettability of the hydrophobic fiber (PA66), enhancing the interaction between the fiber and the antibiotic [43]. This point should be considered in future work, as wettability studies can be performed to understand how to obtain an improved photocatalyst performance at the first cycle.

**Table 3 polymers-16-02960-t003:** Literature studies regarding the photocatalytic degradation of AMX and SDZ.

Antibiotic	[Antibiotic] (mg L^−1^)	Matrix	Photocatalyst	[Catalyst] (g L^−1^)	k (h^−1^)	Ref.
AMX	5	Ultrapure Water	Ag/TiO_2_(P25)/C_3_N_4_	1.0	3.68	[39]
	50	Distillated water pH 9.0	TiO_2_/ZnO	0.1	0.47	[40]
	100	Distillated water pH 6.7	Chitosan/TiO_2_(P25)	*	0.56 ± 0.01	[44]
	10	PBS pH 8.06	TiO_2_/CQDs/PA66	1.5	0.35 ± 0.01	This work
	10	River water	TiO_2_/CQDs/PA66	1.5	0.197 ± 0.005	This work
SDZ	10	PBS pH 8.3	TiO_2_/CQDs	0.5	0.71 ± 0.02	[28]
	10	PBS pH 8.6	TiO_2_(P25)/CQDs	0.5	4.81 ± 0.06	[19]
	10	*	Magnetic biochar/TiO_2_	0.1	0.124 ± 0.008	[45]
	10	PBS pH 8.06	TiO_2_/CQDs/PA66	1.5	0.37 ± 0.01	This work
	10	River water	TiO_2_/CQDs/PA66	1.5	0.52 ± 0.02	This work

* no information available.

The results here presented evidence the efficiency and stability of the PA66/TiO_2_/CQDs photocatalyst and the potential to be applied to remove antibiotics from water environments. Comparatively to the literature (Table 3), for AMX, both Gao et al. [39] and Thi et al. [40] performed reuse cycles for antibiotic removal. On one hand, Gao et al. tested until the 4th cycle with a gradual small loss of efficiency at each cycle, concluding that the photocatalyst maintained high catalytic activity after four cycles and, therefore, has excellent stability in visible light photodegradation reactions. On the other hand, Thi et al. also tested the photocatalyst for four cycles of reutilization. They washed and performed a thermic treatment of the photocatalyst after each cycle, and verified a small drop (5%) in AMX mineralization efficiency. Thus, the authors conclude that the photocatalyst used maintained its stability and catalytic activity for the removal of AMX from wastewater. For SDZ, only Louros et al. [45] performed the recovery and reuse of the materials up until the 5th cycle. In this case, the authors found no significant differences between *t*_1/2_ obtained for the successive cycles, so the results showed after-use recovery photocatalysts, excellent photostability, and reusability of this photocatalyst.

The PA66/TiO_2_/CQDs photocatalyst developed in this study presented an average performance when compared with others found in the literature. However, these nanofibers have the potential to be easily reused, allowing their use as sustainable photocatalysts for water treatment and contributing to water circularity.

## 4. Conclusions

The TiO_2_/CQDs composite was successfully incorporated into polyamide 66 (PA66) fibers by electrospinning, using a nanoparticles-to-polymer ratio of 1:2. The characterization of the resulting composite (PA66/TiO_2_/CQDs) confirmed the presence of TiO_2_/CQDs in its composition. Meanwhile, the degradation studies demonstrated that PA66/TiO_2_/CQDs improved the removal of AMX and SDZ under solar irradiation, in PBS pH 8.06 and river water. Under the presence of the composite, the *t*_1/2_ of AMX was reduced 30 and 17 times in PBS and river water, respectively, when compared with the degradation in the absence of the photocatalyst. For SDZ, the *t*_1/2_ was 3 and 4 times shorter for PBS and river water, respectively, compared with the photolysis studies. Furthermore, the PA66/ TiO_2_/CQDs could be reused at least in three consecutive cycles of the photodegradation of SDZ in river water, consistently achieving a degradation of up to 88% after 4 h. Therefore, PA66/ TiO_2_/CQDs nanofibers were demonstrated to be efficient photocatalysts under solar irradiation and easy to reuse, having high potential to be used in the removal of antibiotics from aquatic environments. In future work, studies regarding the wettability of hydrophobic fiber should be performed to increase the potential efficiency of these systems in photocatalytic applications. Moreover, future studies should encompass the photodegradation mechanism, the identification of the byproducts, and their toxicity.

## Figures and Tables

**Figure 1 polymers-16-02960-f001:**
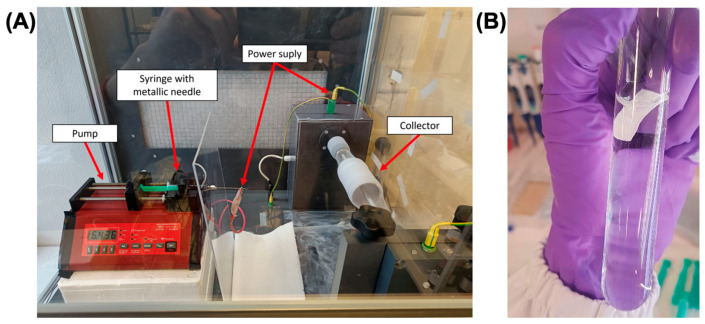
(**A**) System used for the production of nanofibers (E-fiber EF 100). (**B**) Photocatalytic experiment of AMX solution in PBS containing the PA66/TiO_2_/CQDs nanofibers.

**Figure 2 polymers-16-02960-f002:**
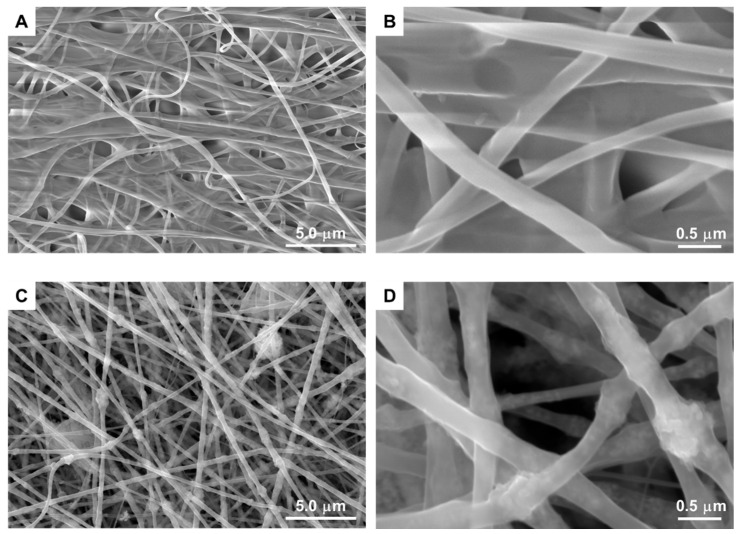
SEM images of PA66 (**A**,**B**) and PA66/TiO_2_/CQDs (**C**,**D**) nanofibers.

**Figure 3 polymers-16-02960-f003:**
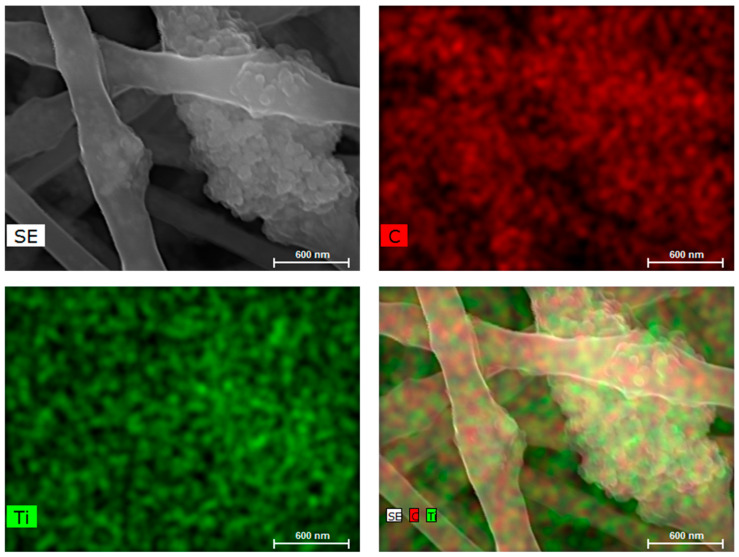
EDS mapping of PA66/TiO_2_/CQDs nanofibers, confirming the presence of C and Ti.

**Figure 4 polymers-16-02960-f004:**
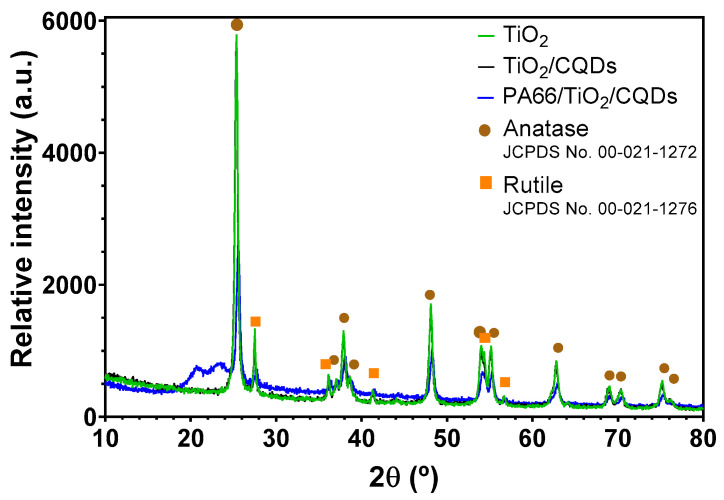
X-Ray diffraction pattern of TiO_2_, TiO_2_/CQDs and PA66/TiO_2_/CQDs composites.

**Figure 5 polymers-16-02960-f005:**
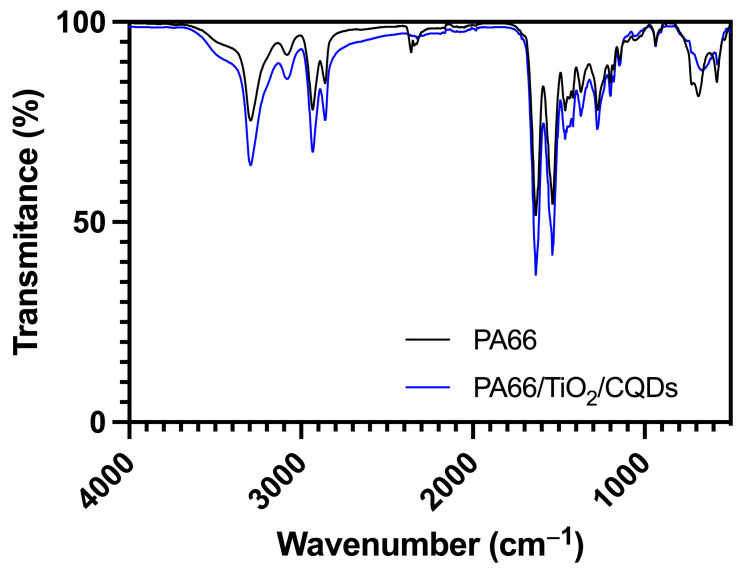
FTIR-ATR of PA66 and PA66/TiO_2_/CQDs nanofibers.

**Figure 6 polymers-16-02960-f006:**
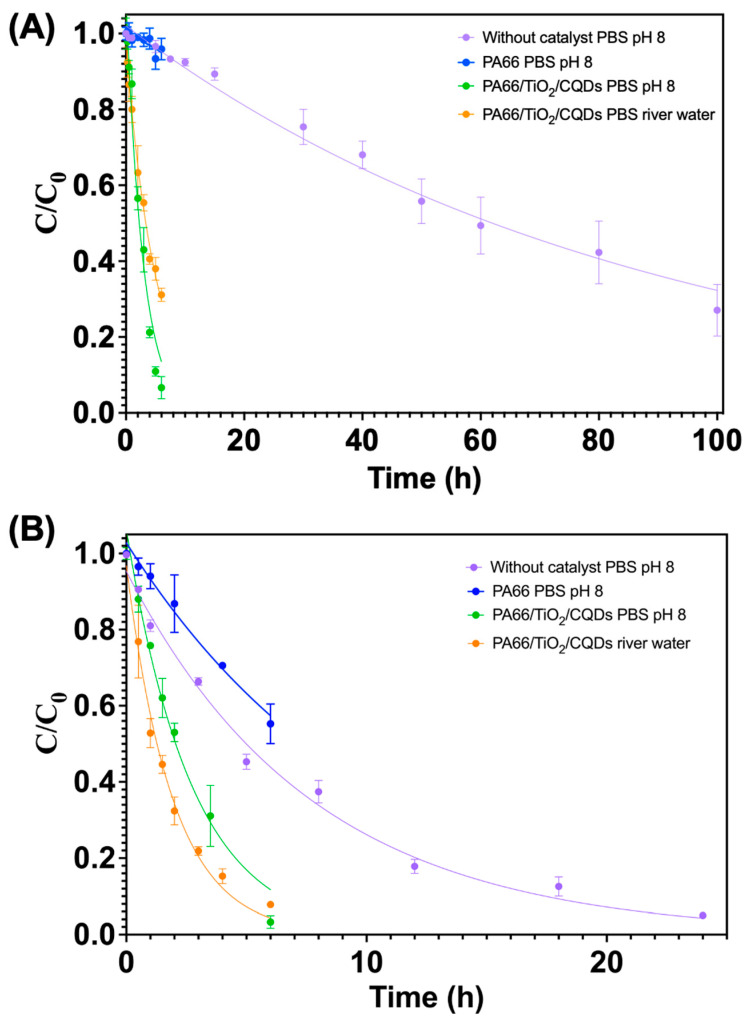
Photodegradation kinetic experimental data together with pseudo-first-order fittings for the photodegradation of (**A**) AMX and (**B**) SDZ (10 mg L^−1^), in the absence of the photocatalyst (photolysis), in the presence of PA66 fibers (1.5 g L^−1^), and with the photocatalyst (PA66/TiO_2_/CQDs, 1.5 g L^−1^), in PBS pH 8.06 or river water. The error bar represents the standard deviation (n = 3).

**Figure 7 polymers-16-02960-f007:**
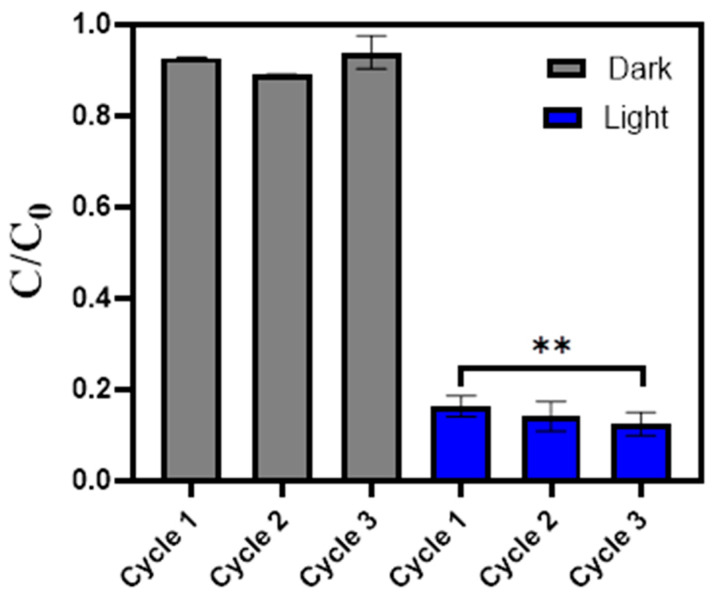
Reuse cycles of PA66/TiO_2_/CQDs (1.5 g L^−1^) in the photodegradation of SDZ (10 mg L^−1^), in the absence of light (Dark, grey bars) and with irradiation for 4 h (Light, blue bars), in river water. The error bar represents the standard deviation (*n* = 3) with one-way ANOVA analysis (** *p* < 0.05).

**Table 1 polymers-16-02960-t001:** Characterization values of the water matrices used.

Matrix	pH	Conductivity (µS cm^−1^)	Salinity (PSU)	TDS (mg L^−1^)	DO (%)	TOC (mg L^−1^)
PBS	8.06 ± 0.01	105	0.05	53	88	1.81 ± 0.02
RIVER WATER	7.08 ± 0.01	171	0.1	101	120.5	4.8 ± 0.1

**Table 2 polymers-16-02960-t002:** Fitting of the experimental data to the pseudo-first-order kinetic model, including pseudo-first-order rate constants (*k* (h^−1^)), determination coefficient (*R*^2^), and half-life times (*t*_1/2_ (h)) obtained for the photodegradation of 10 mg L^−1^ of antibiotic in PBS (pH 8.06) and river water in absence and presence of the photocatalyst. Note: SD is the standard deviation (n = 3).

Antibiotic	Matrix	Catalyst	k ± SD (h^−1^)	*R* ^2^	*t*_1/2_ (h)
AMX	PBS	n.a.	0.012 ± 0.002	0.9572	60 ± 1
	PBS	PA66/TiO_2_/CQDs	0.35 ± 0.01	0.9625	1.98 ± 0.06
	River water	PA66/TiO_2_/CQDs	0.197 ± 0.005	0.9702	3.52 ± 0.09
SDZ	PBS	n.a.	0.129 ± 0.003	0.9887	5.4 ± 0.1
	PBS	PA66/TiO_2_/CQDs	0.37 ± 0.01	0.9649	1.87 ± 0.05
	River water	PA66/TiO_2_/CQDs	0.52 ± 0.02	0.9641	1.33 ± 0.05

n.a.: non-applicable.

## Data Availability

Data are contained within the article and Appendix A.

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
