# Peer review of "Electrospun Nanofiber Dopped with TiO2 and Carbon Quantum Dots for the Photocatalytic Degradation of Antibiotics"

_polymers, 2024, doi:10.3390/polym16212960_

Round 1
Reviewer 1 Report
Comments and Suggestions for Authors
The authors meticulously detail the synthesis and characterization of a novel photocatalyst composed of polyamide 66 (PA66) nanofibers doped with titanium dioxide (TiO2) and carbon quantum dots (CQDs) for the degradation of amoxicillin (AMX) and sulfadiazine (SDZ) under simulated solar radiation. Therefore, I recommend this manuscript for publication after minor revisions addressing the aforementioned points. My comments on the present manuscript are given below.
1) Authors should provide more details on the photocatalytic mechanism of AMX and SDZ by PA66/TiO2/CQDs as catalyst, including the identification of possible byproducts and evaluation of their toxicity.
2) Due to the high adsorption of SDZ identified in the dark phase of catalytic reuse cycles (Figure 7). I recommend that authors determining the adsorption isotherm parameters of this PA66/TiO2/CQDs catalyst.
3) I believe these studies may help in this work and I recommend adding these references in this manuscript.
Author Response
Response to Reviewers Comments
Reviewer 1
Reviewer’s Evaluation
The authors meticulously detail the synthesis and characterization of a novel photocatalyst composed of polyamide 66 (PA66) nanofibers doped with titanium dioxide (TiO2) and carbon quantum dots (CQDs) for the degradation of amoxicillin (AMX) and sulfadiazine (SDZ) under simulated solar radiation. Therefore, I recommend this manuscript for publication after minor revisions addressing the aforementioned points. My comments on the present manuscript are given below.
Thank you for taking the time to review this manuscript. Please find the detailed responses below and the corresponding revisions/corrections highlighted/in track changes in the re-submitted files.
Point-by-point response to Comments and Suggestions for Authors
Comment 1: Authors should provide more details on the photocatalytic mechanism of AMX and SDZ by PA66/TiO2/CQDs as catalyst, including the identification of possible byproducts and evaluation of their toxicity.
Response 1: The authors thank the reviewer for pointing this out. The present work focuses on the photocatalysts’ efficiency in the removal of the two antibiotics, AMX and SDZ, from different matrices and the proof of concept of reusability of this type of photocatalysts (fiber-based). In fact, the work did not include nether the photocatalytic mechanism of any antibiotic, the byproducts identification nor byproducts toxicity. However, authors admit the importance of these studies and a paragraph was added to the manuscript, discussing the possible mechanism based on the data found in the literature. Moreover, the importance of future studies regarding the mechanism of PA66/TiO2/CQDs as photocatalyst, photodegradation byproducts, and toxicity, was also added to the text.
Page 11 line 377: “When TiO2-based composites are used, the photocatalytic process initiates with the absorption of a photon by the semiconductor, exciting an electron (e-) to the conduction band and generating a hole (h+) in the valence band. These two charged species can recombine and transfer heat, or participate in redox reactions. From the reports found in the literature, photocatalysis with TiO2-based composites occurred mainly by redox reactions with water molecules, forming reactive oxygen species (ROS). The formed ROS will then react with the organic pollutants, degrading them [41,42]. Given the efficiency of the evaluated composites, future studies on the specific mechanism of the photocatalytic degradation of AMX and SDZ by this photocatalyst should be conducted, including the evaluation of photodegradation byproducts and their toxicity.”
Comment 2: Due to the high adsorption of SDZ identified in the dark phase of catalytic reuse cycles (Figure 7). I recommend that authors determining the adsorption isotherm parameters of this PA66/TiO2/CQDs catalyst.
Response 2: The authors acknowledge that Figure 7 presents C/C0 instead of SDZ removal. These magnitudes are inversely related, therefore, Figure 7 shows small adsorption of SDZ in the dark tests (less than 15%), and a high efficiency of SDZ removal in the photocatalytic experiments.
Comments 3: I believe these studies may help in this work and I recommend adding these references in this manuscript.
https://doi.org/10.1016/j.mtsust.2024.100937
Response 3: The authors acknowledge the importance of this reference, which was included in the in introduction (line 62).

Reviewer 2 Report
Comments and Suggestions for Authors
I accept the manuscript to be published after minor revision. synthesis of novel photocatalyst based on TiO2/carbon quantum dots is interesting, however, major revision is required of work
1-The novelty of work must be described in the introduction rather than the previous work based on titania/carbon quantum dots
2-TEM analysis is required for verification the preparation of carbon quantum
3-The references required to be updated
4-The photocatalytic mechanism required to be explained
5-The crystalline planes must inserted in XRD figure with card number
6-The crystalline size can be calculated from XRD data using Scherrer equation
7-DRS analysis is required to investigate the effect of carbon quantum dots on the band gap energy of titania
8-XRD of pure titania only is required in the revised manuscript
9-More details on the photocatalytic experiment is required in the revised manuscript
Comments on the Quality of English LanguageThe English language is appropriate
Author Response
Reviewer 2
Reviewer’s Evaluation
I accept the manuscript to be published after minor revision. synthesis of novel photocatalyst based on TiO2/carbon quantum dots is interesting, however, major revision is required of work.
Thank you for taking the time to review this manuscript. Please find the detailed responses below and the corresponding revisions/corrections highlighted/in track changes in the re-submitted files.
Point-by-point response to Comments and Suggestions for Authors
Comment 1: The novelty of work must be described in the introduction rather than the previous work based on titania/carbon quantum dots.
Response 1: The authors thank the revisor for pointing this out. In sequence, we changed the introduction section, as follows:
Page 02, line 91: “In this work, a new photocatalyst was prepared by incorporating TiO2/CQDs composite in polyamide 66 (PA66) nanofibers by electrospinning, and applied in the removal of antibiotics from different aqueous matrices. In previous studies, TiO2/CQDs composites showed excellent photocatalytic efficiency, under solar light irradiation, for the degradation of antibiotics from water [19,28], but evidencing some challenges concerning their reusability. Electrospun PA66 nanofibers were selected for this work to be combined with TiO2/CQDs due to their insolubility in water, good chemical and thermal resistance, high mechanical strength, and low cost [29,30], allowing to obtain a novel and reusable photocatalyst which presented great efficiency under solar light irradiation. Herein, we demonstrated the photocatalytic efficiency of PA66/TiO2/CQDs nanofibers in removing two antibiotics, AMX and SDZ, from different water matrices. Furthermore, as proof of concept, the recovery and reusability of the polymeric photocatalyst were evaluated in river water for SDZ.”
Comments 2: TEM analysis is required for verification the preparation of carbon quantum dots.
Response 2: A TEM image of CQDs was added to the supplementary material file.
Comments 3: The references required to be updated.
Response 3: The authors followed the reviewer’s advice and improved the content of references/literature. Now, in the reviewed manuscript, 44 of the 45 references are from the last 10 years, and 75 % of the cited works are from the last 5 years.
Comments 4: The photocatalytic mechanism required to be explained.
Response 4: The authors thank the reviewer for pointing this out. The present work focuses on the photocatalysts’ efficiency in the removal of the two antibiotics, AMX and SDZ, from different matrices and the proof of concept of reusability of this type of photocatalysts (fiber-based). In fact, the work did not include nether the photocatalytic mechanism of any antibiotic, the byproducts identification nor byproducts toxicity. However, authors admit the importance of these studies and a paragraph was added to the manuscript, discussing the possible mechanism based on the data found in the literature. Moreover, the importance of future studies regarding the mechanism of PA66/TiO2/CQDs as photocatalyst, photodegradation byproducts, and toxicity, was also added to the text.
Page 11 line 393: “When TiO2-based composites are used, the photocatalytic process initiates with the absorption of a photon by the semiconductor, exciting an electron (e-) to the conduction band and generating a hole (h+) in the valence band. These two charged species can recombine and transfer heat, or participate in redox reactions. From the reports found in the literature, photocatalysis with TiO2-based composites occurred mainly by redox reactions with water molecules, forming reactive oxygen species (ROS). The formed ROS will then react with the organic pollutants, degrading them [41,42]. Given the efficiency of the evaluated composites, future studies on the specific mechanism of the photocatalytic degradation of AMX and SDZ by this photocatalyst should be conducted, including the evaluation of photodegradation byproducts and their toxicity.”
Comments 5: The crystalline planes must inserted in XRD figure with card number.
Response 5: The card number was added to the XRD figure.
Comments 6: The crystalline size can be calculated from XRD data using Scherrer equation.
Response 6: The authors acknowledge the review for noticing that this data was missing in the manuscript. Therefore, this information was added in the revised version of the manuscript.
Page 7, line 286: “Moreover, the crystalline size of TiO2 in the PA66/TiO2/CQDs nanofibers was estimated as 20.6 nm, by the Scherrer equation, which similar to bare TiO2 (19.0 nm) and to TiO2/CQDs composites (20.3 nm).”
Comments 7: DRS analysis is required to investigate the effect of carbon quantum dots on the band gap energy of titania.
Response 7: The effect of the CQD on the band gap energy of TiO2 was studied elsewhere (Silva et al. (2022)). This information was added to the manuscript. The value of band gap of both TiO2 and TiO2/CQDs was determined to be 3.2 eV and 3.0 eV, respectively.
Page 06, line 251: “The TiO2/CQDs composites here presented have already shown great results in photocatalytic studies using solar radiation. In a previous work, the authors observed that the association of CQDs (Figure S1) with TiO2 P25 decreased the semiconductor band gap. For this composite, in previous studies, it was observed a change from 3.2 to 3.0, for TiO2 and TiO2/CQDs, respectively [31].”
Comments 8: XRD of pure titania only is required in the revised manuscript.
Response 8: The authors thank the reviewer for pointing this out. The XRD of TiO2 was added to the Figure 4 of the manuscript.
Comments 9: More details on the photocatalytic experiment is required in the revised manuscript.
Response 9: The authors thank the reviewer for pointing this out. The description of photocatalytic experiments was revised and more details were added.
Page 05, lines 194: “For the kinetic photodegradation curves without the catalyst (photolysis), antibiotics aqueous solutions (20 mL, 10 mg L-1) were added to the quartz tubes (internal diameter × height = 1.8 × 20 cm), in triplicate, and irradiated at different times. Aliquots (1.0 mL) of replicates and dark controls were withdrawn at pre-set irradiation times (t, h), stored in the dark at 4 °C and analysed within 24 h for the antibiotic concentration (details are given in section 2.7). Dark controls correspond to a quartz tube covered by aluminium foil maintained inside the solar simulator during the same periods as the irradiated samples, being exposed to identical conditions except for the irradiation.
As for the photocatalytic experiments, the irradiation was carried out under the same conditions, but in the presence of the photocatalyst (PA66/TiO2/CQDs, 1.5 g L-1, Figure 1B). For this, approximately 30 mg of the nanofiber was cut and added to the antibiotics aqueous solutions (10 mg L-1) in quartz tubes (internal diameter × height = 1.8 × 20 cm), in triplicate, and the mixture was irradiated at different times. The volume of the antibiotic was calculated considering the fiber mass, in order to have a 1.5 g L-1 of the photocatalyst.”
Page 05, lines 216: “For the reuse experiments, the nanofiber was added to SDZ in river water (10 mg L-1), maintaining the photocatalyst dose at 1.5 g L-1, in quartz tubes, and irradiated for 4 h. Then, the antibiotic solution was removed from the quartz tube, and the same volume of a new SDZ solution was added. This procedure was repeated to perform three consecutive cycles. Each cycle was done in triplicate, and dark controls were also evaluated. Each sample was filtered and analysed according to section 2.7.”

Reviewer 3 Report
Comments and Suggestions for Authors
The paper reported on a polymer-immobilized version of a photocatalyst based on carbon quantum dots at titanium dioxide. This catalyst was developed for the purpose of wastewater treatment, with a focus on removing antibiotics. T. The catalyst was tested the on two specific compounds: amoxicillin (AMX) and sulfadiazine (SDZ). The catalyst demonstrated average performance in tests, and its advantage could be its potential for reuse. However, due to the small number of reuse cycles tested, it is hard to conclude about the long-term sustainability of the catalyst. Additionally, the complete removal of SDZ during the reuse was not demonstrated, and the tests were limited to only SDZ. It is unclear why AMX was not included in the reuse tests, as it would have provided more information about the catalyst's overall performance. Given these limitations, the conclusions drawn from the study are not fully supported and may be incorrect. For example, the statement that the results provide evidence of the efficacy and stability of the photocatalyst is not supported by the data presented. The same applies to the following statements: The nanofibers were demonstrated to be efficient photocatalysts under solar irradiation and easy to reuse, having high potential to be used in the removal of antibiotics from aquatic environments.
Another point of concern is the increasing of the efficiency upon reuse. There should be some explanation or maybe the experimental error is too large?
The authors reported that the bare fiber PA66 exerts a filter effect that inhibits the photodegradation of SDZ, but the removal of AMX was not impacted by the polymer. The explanation of that should be provided.
SDZ removal was faster in river water than in PBS, and authors claim that it is in accordance with literature where the same behavior was observed. However, the cited paper reported that a higher pH accelerated photodegradation,. However, this is not the case here, as the river water has a lower pH and a faster reaction rate. Lower pH and lower rate was in AMX degradation. In order to make a correct comparison between river water and the buffer, it would be better to maintain the same pH and salinity in both water matrixes. The authors should have considered this when choosing the matrixes.
Overall, the paper needs a major revision including additional experiments and can be accepted upon addressing the above points.
Author Response
Reviewer 3
Reviewer’s Evaluation
The paper reported on a polymer-immobilized version of a photocatalyst based on carbon quantum dots at titanium dioxide. This catalyst was developed for the purpose of wastewater treatment, with a focus on removing antibiotics. The catalyst was tested the on two specific compounds: amoxicillin (AMX) and sulfadiazine (SDZ)…
Overall, the paper needs a major revision including additional experiments and can be accepted upon addressing the above points.
Thank you for taking the time to review this manuscript. Please find the detailed responses below and the corresponding revisions/corrections highlighted/in track changes in the re-submitted files.
Point-by-point response to Comments and Suggestions for Authors
Comments 1: The paper reported on a polymer-immobilized version of a photocatalyst based on carbon quantum dots at titanium dioxide. This catalyst was developed for the purpose of wastewater treatment, with a focus on removing antibiotics. The catalyst was tested the on two specific compounds: amoxicillin (AMX) and sulfadiazine (SDZ). The catalyst demonstrated average performance in tests, and its advantage could be its potential for reuse. However, due to the small number of reuse cycles tested, it is hard to conclude about the long-term sustainability of the catalyst. Additionally, the complete removal of SDZ during the reuse was not demonstrated, and the tests were limited to only SDZ. It is unclear why AMX was not included in the reuse tests, as it would have provided more information about the catalyst's overall performance. Given these limitations, the conclusions drawn from the study are not fully supported and may be incorrect. For example, the statement that the results provide evidence of the efficacy and stability of the photocatalyst is not supported by the data presented. The same applies to the following statements: The nanofibers were demonstrated to be efficient photocatalysts under solar irradiation and easy to reuse, having high potential to be used in the removal of antibiotics from aquatic environments.
Response 1: The authors acknowledge that a small number of reuse cycles was performed and only for one condition (SDZ in river water). This section of the work was performed as a proof of concept for the application of the developed photocatalyst, therefore, only one condition was tested. Furthermore, to follow the removal of the antibiotic SDZ by the photocatalyst, and determine if the same efficiency was maintained in the reuse cycles, it was decided to select conditions to achieve a degradation of around 85% (corresponding to 4 h of irradiation), in order to observe a significant removal of SDZ but still corresponding to a measurable concentration of the antibiotic (above the LOQ of the analytical method). Moreover, over the last 5 years, many literature studies have performed only 3 cycles as proof of the potential of the photocatalyst (references below), like in this work.
References:
- https://doi.org/10.1016/j.cej.2018.10.186
- https://doi.org/10.1016/j.jallcom.2020.156077
- https://doi.org/10.1016/j.jenvman.2020.110839
- https://doi.org/10.1016/j.jphotochem.2020.112976
- https://doi.org/10.1016/j.cej.2021.131594
- https://doi.org/10.1016/j.seppur.2022.122205
- https://doi.org/10.1016/j.envres.2023.116246
- https://doi.org/10.1016/j.jenvman.2022.116396
- https://doi.org/10.1016/j.seppur.2023.125245
- https://doi.org/10.1016/j.matchemphys.2024.129300
- https://doi.org/10.1016/j.jphotochem.2023.115154
Comments 2: Another point of concern is the increasing of the efficiency upon reuse. There should be some explanation or maybe the experimental error is too large?
Response 2: The authors thank the reviewer for pointing out this information. In fact, there is a slight increase of the removal of SDZ during the reuse cycles. To verify if the increase is significant, a one-way ANOVA statistical test was performed and included in the manuscript.
Page 05, lines 227: “GraphPad Prism 8 was used to apply a one-way ANOVA to compare the photocatalytic efficiencies of each reuse cycle.”
The one-way ANOVA revealed that only cycle 1 and cycle 3 of photodegradation is significantly different. The adsorption between cycles is not significantly different nether the results of photodegradation between cycle 1 and cycle 2 nor cycle 2 and cycle 3. The increase in efficiency of the photocatalyst between cycle 1 and cycle 3 may be due to the increase in the wettability of the hydrophobic fiber: due to the continuous presence of the membrane in water, the chemical–physical properties of its surface might have changed and allowed the solution, and consequently the antibiotic, to interact with active sites not available in previous cycles. This point should be considered in future work, as wettability studies can be performed to understand how to obtain improved photocatalyst performance at the first cycle.
Page 11, line 405: “The results showed a slight enhancement in SDZ removal until the 3rd cycle of reuse, increasing from 84% (1st cycle) to 88% (3rd cycle). One-way ANOVA revealed that only the photocatalytic results corresponding to cycles 1 and 3 are significantly different (p < 0.05), with dark controls presenting no significant difference between the three cycles. The small increase in efficiency of the photocatalyst with the reuse cycles can be due to the increase in the wettability of the hydrophobic fiber (PA66), enhancing the interaction between the fiber and the antibiotic [45]. This point should be considered in future work, as wettability studies can be performed to understand how to obtain improved photocatalyst performance at the first cycle.”
Page 13, lines 459: “In future work, studies regarding the wettability of the hydrophobic fiber should be performed to increase the potential efficiency of these systems in photocatalytic applications.”
Comments 3: The authors reported that the bare fiber PA66 exerts a filter effect that inhibits the photodegradation of SDZ, but the removal of AMX was not impacted by the polymer. The explanation of that should be provided.
Response 3: The authors acknowledge that a comment was missing on the original manuscript about this issue. As happens with many antibiotics, depending on their structure and class, their interaction with the matrix, adsorbents, and photocatalysts differ. Although for SDZ, a sulphonamide, it was verified that the presence of bare fiber PA66 without composite exerts a filter effect, decreasing the SDZ removal from solution comparatively to SDZ photolysis, the same was not verified for AMX. In fact, AMX, an aminopenicillin, removal had similar behaviour with and without bare fiber PA66.
Several parameters can interfere differently with the removal of each antibiotic. For example, Louros et al (2020) identified that SDZ removal is negatively affected by high salinity and slightly by scavenger of ●OH (presence of propan-2-ol). However, for AMX, Rocha et al (2024) identified that the scavenger of ●OH highly impacts this antibiotic removal, decreasing its degradation comparatively to photolysis in PBS (pH 8.05). On the other hand, scavengers of 1O2 highly increase the removal of SDZ but slightly affect the removal of AMX. Therefore, PA66 fiber can act as a filter and inhibit the photodegradation of SDZ while not affecting the removal of AMX.
Page 11, line 356: “As happens with many antibiotics, depending on their structure and class, their interaction with the matrix, adsorbents, and photocatalysts differ. In this work, for SDZ, a sulphonamide, it was verified that the presence of bare fiber PA66 without composite exerts a negative effect in degradation of this antibiotic, decreasing its SDZ removal from solution comparatively to SDZ photolysis. The same was not verified for AMX, an aminopenicillin, whose removal had similar behaviour with and without bare fiber PA66. Thus, for SDZ, the fiber has a filter effect, acting as scavenger of reactive species or quenching the excited states, and affecting the photodegradation process [6,38].”
Comments 4: SDZ removal was faster in river water than in PBS, and authors claim that it is in accordance with literature where the same behavior was observed. However, the cited paper reported that a higher pH accelerated photodegradation. However, this is not the case here, as the river water has a lower pH and a faster reaction rate. Lower pH and lower rate was in AMX degradation. In order to make a correct comparison between river water and the buffer, it would be better to maintain the same pH and salinity in both water matrixes. The authors should have considered this when choosing the matrixes.
Response 4: The photocatalytic efficiency can be affected by pH and the matrix composition. In this case, the authors observed that the degradation of SDZ was faster in river water than in PBS. In the work cited in this manuscript, the photodegradation of SDZ also occurred at a higher rate in freshwater (pH 7.3) than in ultrapure water (pH 7.3), where pH had no influence. In this cited work, the pH interference was evaluated at pH 6.3 and 7.3, and showed that the degradation was slower at lower pH. However, in our case, the pH was not the same, but the matrix where the photodegradation rate was higher was at a lower pH, so it seems that the main factor affecting our degradation rates should be the aqueous matrix, and its organic matter content, which overcomes the negative influence of the pH. In fact, Louros et al. (2020) presented that SDZ removal in freshwater (river water from Aveiro), was much faster compared to any pH studied by the author. Furthermore, Louros et al. (2020) performed this study in the same pH as the river water, revealing that the pH was not the factor that impacted the faster removal of SDZ. Thus, since both PBS and river water presented in this study had low salinity, as can be seen in Table 1, the principal difference that can be identified is the TOC value which is much higher in river water and should be responsible for the effect. As verified by Louros et al (2020), the principal factor that impacted the high removal of SDZ in freshwater was the creation of 3DOM* species that dominated other DOM effects such as back-reductions, inner filter effect, or scavenging/quenching of reactive species. Therefore, the extrapolation of the results herein obtained, in our point of view, can be performed.
To clarify this, the text was changed.
Page 11, lines 3: “SDZ removal was faster in river water than in PBS, with a t1/2 decrease from 1.87 ± 0.05 h to 1.33 ± 0.05 h. In previous works, the photodegradation of SDZ also occurred at a higher rate in freshwater (pH 7.3) than in ultrapure water (pH 7.3), where pH had no influence. In fact, it was reported that the increase in pH from 6.3 to 7.3 has a positive influence on SDZ photodegradation due to a higher prevalence of SDZ in its negatively charged form. In our case, the pH was not the same, however, the matrix where the photodegradation rate was higher was at a lower pH, indicating that the main factor affecting our degradation rate was the aqueous organic matter content, that overcame the negative influence of the pH. Louros et al. (2020) reported that the principal factor that impacted the high removal of SDZ in freshwater was the creation of 3DOM* species.”

Round 2
Reviewer 2 Report
Comments and Suggestions for Authors
I accept the revised manuscript for publication
Comments on the Quality of English LanguageThe English language is well expressed in the manuscript
Reviewer 3 Report
Comments and Suggestions for Authors
The revised version of the manuscript has been sufficiently improved and can now be published in Polymers.